# Metformin: Sentinel of the Epigenetic Landscapes That Underlie Cell Fate and Identity

**DOI:** 10.3390/biom10050780

**Published:** 2020-05-18

**Authors:** Javier A. Menendez

**Affiliations:** 1Program against Cancer Therapeutic Resistance (ProCURE), Metabolism and Cancer Group, Catalan Institute of Oncology, 17007 Girona, Spain; jmenendez@idibgi.org; Tel.: +34-872-987-087; 2Girona Biomedical Research Institute, Salt, 17190 Girona, Spain

**Keywords:** aging, cancer, epigenetics, DNA, histones, methylation

## Abstract

The biguanide metformin is the first drug to be tested as a gerotherapeutic in the clinical trial TAME (Targeting Aging with Metformin). The current consensus is that metformin exerts indirect pleiotropy on core metabolic hallmarks of aging, such as the insulin/insulin-like growth factor 1 and AMP-activated protein kinase/mammalian Target Of Rapamycin signaling pathways, downstream of its primary inhibitory effect on mitochondrial respiratory complex I. Alternatively, but not mutually exclusive, metformin can exert regulatory effects on components of the biologic machinery of aging itself such as chromatin-modifying enzymes. An integrative metabolo-epigenetic outlook supports a new model whereby metformin operates as a guardian of cell identity, capable of retarding cellular aging by preventing the loss of the information-theoretic nature of the epigenome. The ultimate anti-aging mechanism of metformin might involve the global preservation of the epigenome architecture, thereby ensuring cell fate commitment and phenotypic outcomes despite the challenging effects of aging noise. Metformin might therefore inspire the development of new gerotherapeutics capable of preserving the epigenome architecture for cell identity. Such gerotherapeutics should replicate the ability of metformin to halt the erosion of the epigenetic landscape, mitigate the loss of cell fate commitment, delay stochastic/environmental DNA methylation drifts, and alleviate cellular senescence. Yet, it remains a challenge to confirm if regulatory changes in higher-order genomic organizers can connect the capacity of metformin to dynamically regulate the three-dimensional nature of epigenetic landscapes with the 4th dimension, the aging time.

There is increasing awareness that non-genetic, environmental, and metabolic stimuli can modify the structure and function of chromatin as a cause of aging [1,2,3,4,5,6]. Such disruption of the homeostatic resilience of chromatin has the potential to generate ectopic gene expression profiles, deregulate differentiation, and activate maladaptive cell fate programs in a purely epigenetic manner. The epigenetic states underlying the plastic nature of cell fate decisions therefore operate as core elements of a tissue’s capacity to undergo successful repair, aging degeneration or malignant transformation in response to injury, stress, and disease. A definitive understanding of the mechanisms for safeguarding the epigenome architecture for cell identity might radically change not only our conceptual approach to disease mechanisms, but also the way we therapeutically manage aging-related diseases such as cancer [7,8,9,10,11,12,13,14,15].

Current disease models and drug discovery strategies are largely biased towards the prevailing, deterministic mutation theory of most chronic, noninfectious diseases. Such a framework, however, fails to capture the ability of the epigenome to absorb the effects of stochastic perturbations in a purposeful, programmed manner, and to modulate cellular plasticity in development and disease [16,17,18,19]. The aging research field therefore needs to develop, test, and validate a new generation of conceptual, experimental, and therapeutic approaches aimed to molecularly deconstruct and pharmacologically target the epigenetic landscape itself. With regard to the latter, the biguanide derivative metformin—the first-line therapeutic for type 2 diabetes for more than 60 years and the first drug chosen to be tested in a clinical trial aimed to target the biology of aging per se [18,19,20]—may inspire the development of a new generation of anti-aging molecules targeting epigenetic plasticity.

## 1. Metformin: From Metabolic to Epigenetic Offenders of the Aging Process

The stochastic and dynamic nature of chromatin structure and epigenetic information allows cells to reach and maintain a differentiated state. The organizational structure of chromatin also mediates developmental, reparative, or malignant phenotypic plasticity throughout the aging process [16,17,18,19]. The relationship between chromatin structure and its functional sensitivity to (micro)environmental conditions, mediated by epigenetic modifications (e.g., methylation and acetylation), maximizes the organismal efficiency to store epigenetic information. In turn, the capacity of the epigenome to regulate the signal-to-noise ratio and, consequently, to promote or resolve cell-state transitions, and to dictate robustness during cell fate commitment [16,17,18,19], ultimately drives the pathophysiology of aging. In this regard, non-genetic stimuli such as inflammation, hypoxia, cell stress, developmental cues, or metabolism, can promote overly restrictive chromatin states—capable of preventing the induction of tumor suppression programs or blocking normal differentiation—or promote overly permissive chromatin states—capable of stochastically activating oncogenic programs or non-physiological cell fate transitions [2,5]. Within this new framework, how might metformin affect or even manipulate how cells access (or exit) cellular identities in a dynamic epigenetic landscape?

The current consensus is that metformin exerts indirect pleiotropy on core metabolic hallmarks of aging, such as the insulin/insulin-like growth factor 1(IGF-1) and AMP-activated protein kinase (AMPK)/mammalian Target Of Rapamycin (mTOR) signaling pathways, downstream of its primary inhibitory effects on mitochondrial respiratory complex I [20,21] (Figure 1A). Alternatively, but not mutually exclusive, its capacity to operate as an anti-aging agent might involve (directly or indirectly) regulatory interactions with components of the biologic machinery of aging itself such as chromatin-modifying enzymes [22,23,24,25] (Figure 1B). An integrative metabolo-epigenetic outlook supports a new model whereby metformin operates as a guardian of cell identity, capable of retarding cellular aging by preventing the loss of the information-theoretic nature of the epigenome (Figure 1C).

## 2. Metformin: A Metabolic Landscaper of the Epigenome

The substrates and cofactors employed to generate nucleic acid and chromatin modifications (e.g., methylation and acetylation) are affected not only by the availability of nutrients (including glucose, acetate, serine, glycine, threonine, and methionine), but also by the activity of metabolic pathways (e.g., tricarboxylic acid (TCA) cycle and serine-glycine one-carbon metabolism). The epigenome is causally implicated in the establishment and maintenance of cellular states, that is, heritable chromatin modification profiles define a phenotype while alterations to chromatin marks serve as limiting steps (barriers) to cell fate transitions. It follows consequently that metabolic reprogramming could dynamically reshape the epigenomic landscape [26,27,28,29,30,31,32,33]. The biochemical basis for the influence of metabolites on epigenomics is grounded on the fact that the physiological concentrations of the substrates/cofactors involved are commensurate with the kinetic parameter ranges of the chromatin-modifying enzymes (i.e., K_m_, K_i_, and IC_50_), making them responsive to changes in metabolism. Such histone modifiers and chromatin remodelers include histone acetyltransferases (acetyl-CoA and CoA), histone methyltransferases (s-adenosylmethionine (SAM), s-adenosylhomocysteine (SAH), and methylthioadenosine (MTA)), DNA methyltransferases (SAM, SAH, and MTA), histone deacetylases (nicotinamide adenine dinucleotide (NAD^+^), nicotinamide, and β-hydroxybutyrate), histone demethylases (α-ketoglutarate, flavin adenine dinucleotide (FAD), R/S 2-hydroxyglurarate, succinate, fumarate, and FADH_2_), and DNA demethylases (α-ketoglutarate, R/S 2-hydroxyglurarate, succinate, and fumarate).

Using Waddington’s “epigenetic landscape” as a metaphor for epigenotypes (valleys) and transition barriers (summits), one can visualize how metformin might operate as a metabolic caretaker of the epigenome (Figure 2). The capacity of metformin to affect the availability of substrates/cofactors required for specific chromatin modifications could alter the height of phenotypic barriers, thereby facilitating (or impeding) the transition from one cell type to another (e.g., differentiation). Indeed, several examples illustrate the ability of metformin to alter the levels of metabolites related to chromatin remodeling, namely: NAD^+^ (by acting as an inhibitor of mitochondrial NADH (nicotinademide adenine dinucleotide reduced) dehydrogenase and preventing NAD^+^ regeneration [34,35,36,37]), acetyl-CoA (by impeding the acetyl-CoA carboxylase-catalyzed conversion of acetyl-CoA to malonyl-CoA [38]), α-ketoglutarate (by decreasing mitochondrial respiration and TCA cycle activity [39,40,41]), the SAM:SAH ratio (by targeting the coupling between serine mitochondrial one-carbon flux and AMPK-sensed complex I activity) [22,23], or β-hydroxybutyrate (by promoting mitochondrial fatty acid β-oxidation [42,43,44,45]). Since glucose induces histone O-GlcNAcylation (O-N-acetyl glucosaminylation) via the hexosamine biosynthesis pathway [46,47], the ability of metformin to impair glucose consumption by acting as an inhibitor of hexokinase-II [48,49] may globally link the cellular energy status with the regulation of the epigenome [50,51]. Additionally, there are several examples of the metabolic ability of metformin to facilitate (or impede) the transition from one cell type to another, including neuronal [52,53,54], osteogenic [55,56,57,58], adipogenic [39,59], myofibroblast and myoblast [60,61] differentiation, or monocyte-to-macrophage differentiation [62], among others. The fact that metformin targets a set of core “hub” metabolites with epigenetic properties and with emerging roles as central mediators of aging strongly supports the notion that it can regulate cell fate transitions by changing metabolite levels that allow the reorganization of specific chromatin marks [63]. Accordingly, a link between metformin-induced changes in specific metabolites (e.g., α-ketoglutarate and succinate) and the epigenetic control of cell fate transitioning has been established in a few cases [39,41,64]. The ability of metformin to ameliorate the harmful effects of the so-called “metabolic memory” or “legacy effect” (i.e., the long-term persistence of epigenomic alterations and aberrant epiphenotypes despite attaining control of the metabolic trigger [65,66,67,68]) appears to involve coupled changes in epigenetic metabolites, activation status of chromatin-modifying enzymes, and reversal of specific epigenetic marks in command of the epigenetic phenomena (DNA/histone methylation and acetylation) that drive metabolic memory. Nonetheless, forthcoming studies should unambiguously demonstrate that metformin-driven shifts in specific metabolic features could orchestrate specific cell functions through their impact on chromatin dynamics and epigenetic remodeling.

## 3. Metformin: Countering the Erosion of the Epigenetic Landscape

Erosion and consequent “smoothening” of the epigenetic landscape may underlie most (if not all) aspects of aging [69,70,71,72,73,74,75]. A lasting relocation of chromatin modifiers in response to prolonged stress signaling (e.g., aging-related DNA damage) can introduce non-random changes into the epigenome barriers in a manner that allows cell states to aberrantly transit towards other differentiation valleys, consequently leading to a loss of cell fate commitment. Accordingly, dysregulation of developmental genes controlling cellular identity and promoting epigenotype interconversions (e.g., luminal to basal/mesenchymal transdifferentiation of mammary epithelial cells, fibroblasts gaining adipogenic or neuronal traits, myogenic-to-fibrogenic lineage conversion of muscle satellite cells, etc.) has repeatedly been associated with the aging process [76,77,78,79,80,81]. Intriguingly, these loss-of-identity phenomena can be counteracted by well-recognized interventions capable of slowing down aging such as calorie restriction, thereby suggesting that the underlying epigenetic alterations are sensitive to systemic metabolic changes.

Metformin can aid in globally protecting the epigenetic landscape from erosion and preventing the loss of its information-theoretic nature over time (Figure 2 and Figure 3). One of the best scenarios to visualize how metformin can affect and possibly specify cell fate by impeding the reshaping of the epigenetic landscape is the reprogramming of differentiated skin fibroblasts to a pluripotent state [82]. During cell fate conversion processes such as reprogramming, transdifferentiation, or transdetermination, pluripotency transcription factors (or epigenetic noise itself) can blur the boundaries between different cell fates to flatten the hierarchy of cell states [83]. Cellular reprogramming to pluripotency and subsequent differentiation of induced pluripotent stem cells (iPSCs) are accompanied by a rewiring of mitochondrial respiration, mitochondrial network dynamics, and TCA cycle function. Such profound metabolic reprogramming impacts chromatin reorganization and reshapes the epigenome configuration to influence gene expression and change cellular identities [84,85,86,87]. Metformin imposes a metabolic shift away from the metabolic features that fuel pluripotency, thereby endowing fully differentiated somatic cells with a metabolic infrastructure that is protected against epigenetic reprogramming [88]. In fact, the metabolic barriers imposed by metformin cannot be bypassed even through deficiency of cancer-associated mutations such as p53, a fundamental mechanism that greatly improves the efficiency of epigenetic reprogramming of somatic cells to pluripotency [89,90,91]. Conversely, metformin has the ability to suppress the tumorigenic fate of teratoma-initiating cells perversely locked in an undifferentiated, pluripotent state while preventing (and perhaps actively channeling) the terminal differentiation of pluripotent cells into multiple lineages [92,93].

Viewed in a broader context, metformin as an impediment to aberrant cell fate conversion might exemplify its ability to take control of evolutionarily conserved, metabolic reprogramming barriers aimed to prevent the induction of ectopic gene expression programs. Considering a model of metabolism-responsive loss of epigenetic resilience as a mechanism of cellular aging, metformin might impact the stochastic translation of metabolic inputs into resilient/plastic cell states via epigenetic regulatory systems [13,14,15]. Such an unanticipated ability of metformin to preserve the epigenomic landscape is supported by its generally maligned ability to regulate ancient regulatory mechanisms of metabolic physiology such as the so-called non-coding RNAs [94,95,96,97]. Metformin targets the auto-regulatory loop of Lin28/let-7 microRNAs (miRNAs) [98,99,100,101,102], in which Lin28 operates as a gatekeeper of the pluripotent state that binds to and inhibits the processing of let-7, which counteracts the activity of stemness factors by promoting the expression of prodifferentiation genes. The Lin28/let-7 axis links the functional status of mitochondria with the regulation of one-carbon metabolism, nucleotide metabolism, and histone methylation [94,95,96,97]. As such, it is a key component of the positive feedback loop underlying an inflammation-driven epigenetic switch from a nontransformed to a self-renewing transformed cell type without requiring mutational changes [102,103]. These epigenetically driven processes of cellular transformation cannot occur when metformin is present but can be bypassed upon concomitant overexpression of Lin28 [104,105,106]. Metformin could therefore promote differentiation-primed epistates and/or prevent transit into differentiation-refractory (stem-like) state by broadly modulating miRNAs. Accordingly, metformin can control miRNAs by transcriptionally activating the promoter of the RNAse III DICER [107,108,109,110], a key miRNA processing enzyme that controls the balance between transcriptionally favorable and unfavorable histone modifications that affect the expression profiles of developmental genes [111,112,113,114], to ultimately favor cell differentiation. The ability of metformin to target a key controller of miRNA biogenesis might represent a highly effective strategy for fine-tuning how external (environmental) noise alters the cell and tissue homeostasis by epigenetically targeting multiple genes simultaneously.

## 4. Metformin: Remolding and Preserving the Epigenetic Landscape

Chronic stresses can elicit heritable changes in the epigenetic landscape that “lock” cells in abnormal states, leading to aging and aging-related diseases. Such epigenetic changes include a decrease in the abundance of histone H3 and H4, a global loss of heterochromatin marks (e.g., H3K9me3 and H3K27me3), as well as an overall decrease in the activity of sirtuins (SIRT1/SIRT6) and in DNA methylation at CpG sites, both allowing the ectopic transcription of the so-called Long Interspersed Nuclear Elements (LINE) [2,69,71,72,115,116,117,118,119,120,121,122,123,124,125]. Ultimately, when the accumulative shift away from the original epigenetic landscape reaches a critical level, cells might enter senescence if DNA damage-response signaling is constitutively activated [73,74,126]. Metformin can drive phenotypes of extended healthspan and cancer resistance by promoting more resilient epigenomes capable of countering aging-related loss of cell fate and dedifferentiation via heterochromatin preservation and silenced LINE-1 retrotransposons.

Naturally long-lived and cancer-resistant organisms have provided key clues to suggest that engineering more stable, H3K27me3-enriched epigenomes might extend human healthspan and prevent cancer [127], which is consistent with the so-called heterochromatin loss model of aging where there is a trend for increases in activating histone marks (e.g., H3K4m2/3 and H3K26m3) and decreases in repressive histone marks (e.g., H3K9me2/3 and H3K27me3) [2]. Metformin restores the global levels of H3K27me3 in fibroblasts of aged individuals or from patients with premature aging syndromes such as Hutchinson-Gilford progeria or Werner syndrome, likely by directly targeting the H3K27me3 demethylase KDM6A/UTX [24]. A decrease in repression-associated H3K27me3 (and an increased activity of KDM6A/UTX) is a key feature of the global chromatin reconfiguration that takes place not only in somatic cells during physiological aging but also in the prematurely aging cells from patients with Hutchinson-Gilford progeria or Werner syndrome [63,116,117]. Gain of H3K27me3 (and loss of KDM6A/UTX activity) has been linked to extended longevity in *Caenorhabditis elegans*, thus suggesting that metformin-driven preservation of high levels of H3K27me3 by inhibiting KDM6A/UTX may be critical for maintaining a youthful epigenetic landscape [63,128,129,130].

Metformin might contribute to (epi)genome integrity by regulating the activation status of retrotransposable elements such as LINE-1, whose methylation-regulated motility can be viewed as an epigenetic biomarker of aging. Methylation of CpGs in the LINE-1 promoter silences LINE-1 expression, whereas the absence of DNA methylation is permissive for LINE-1 transcription in differentiated somatic cells. LINE-1 elements are therefore typically heavily methylated in the majority of normal young tissues but are subjected to deep epigenetic alterations during aging that culminate in active transposition [121,122,124,131]. Active LINE-1 retrotransposition works via a “copy-and-paste” process that involves LINE-1 DNA endonuclease activity to nick target DNA at the site of insertion. Consequently, the development of LINE-1 hypomethylation in the course of aging may constitute a crucial determinant of loss of genomic integrity and deterioration of the dynamics and plasticity of the human genome. Indeed, de-silencing of retrotransposons such as LINE-1 is molecularly equivalent to accumulating epigenetic noise and erosion of the epigenetic landscape in aging and aging-related diseases [132,133]. Mechanistically, metformin might have the ability to promote genome-wide alterations in the LINE-1-related DNA methylome by at least three complementary mechanisms: First, metformin protects the AMPK-mediated phosphorylation of serine 99 at the Ten-eleven translocation 2 (TET2) demethylating enzyme, which increases TET2 stability and 5-hydroxymethylcytosine (5hmC) levels [134]. Although TET binding drives DNA demethylation and promotes the activity of young LINE-1 elements in the epigenetic landscape of pluripotent stem cells, TET-dependent repressive activities appear to constitute a host defense strategy for ensuring LINE-1 silencing upon TET-mediated DNA demethylation to maintain genome stability [135,136,137]. By ensuring that LINE-1 expression is kept under control via TET2, metformin might impede the ability of pernicious metabolic environments (e.g., high glucose states) to reprogram the epigenome. Second, metformin can increase the SAM:SAH ratio-related capacity of DNA methyltransferases (DNMTs) to methylate LINE-1 [138,139]. On the one hand, metformin-driven activation of AMPK positively modulates the activity of S-adenosylhomocysteine (SAH) hydrolase, the sole mammalian enzyme capable of hydrolyzing SAH, a potent feedback inhibitor of S-adenosylmethionine (SAM)-dependent methyltransferases including DNMTs [22,140,141]. On the other hand, metformin can promote the accumulation of tetrahydrofolate forms carrying activated one-carbon units, which can be critical to boost the level of SAM and SAM-related methylation [142,143]. Moreover, metformin reprograms the store of methylation units by targeting the coupling between serine mitochondrial one-carbon flux and mitochondrial complex I activity and their channeling to affect the methylation status of DNA [23,144]. By increasing the contribution of one-carbon units to the SAM from folate stores while decreasing SAH (and stabilizing TET2) in response to AMPK-sensed energetic crisis, metformin can provide a robust regulatory axis of the global (LINE-1) DNA methylome. Third, NAD^+^–dependent deacetylases such as SIRT1, SIRT6, and SIRT7, a class of lysine deacylases that regulate cellular metabolism and energy homeostasis, are negative controllers of the self-propagation of LINE-1 in the genome [118,122,145,146,147]. Metformin stimulates the activity/expression of SIRT1 and SIRT6 [148,149,150,151], indirectly downstream of AMPK activation but also by directly improving the catalytic efficiency of sirtuins operating in conditions of low NAD^+^ that accompany the aging process [151,152,153,154,155], thereby providing an additional mechanism through which metformin can restrict the dysfunctional activity of LINE-1 during aging.

It is noteworthy that LINEs are enriched at lamina-associated domains (LADs), heterochromatic regions of the nuclear periphery. Such regions should be viewed as dynamically infolding 3D chromatin structures that, by changing nuclear structure in response to intra- and extracellular cues and signals, can alter the depth of the Waddingtonian hills and valleys during differentiation and reprogramming [16,17,18,156,157,158]. The promotion of the anchoring of LINE-1 to the nuclear envelope via enhanced methylation might provide a direct link between metabolo-epigenetic activities of metformin, nuclear lamina, 3D configuration and loci in the nucleus, and epitranscriptional preservation of the epigenetic landscape (Figure 3). Using an epigenetically primed BRCA1 model of cancer initiation that is accompanied by global hypomethylation and phenotypic transdifferentiation due to chronically inadequate DNA damage repair [23,75,159,160,161,162,163,164], metformin treatment was found to enhance global DNA methylation including in LINE-1 retrotransposon sites. Since activation of endogenous LINE-1s and their relocation toward the nuclear interior evolves progressively during cell senescence and involves large scale reorganization of LAD chromatin structure including regions of H3K27me3 marks [69,124,147], forthcoming studies should evaluate whether metformin treatment suffices to promote the recruitment of LINE-1 elements to the nuclear periphery and consequently generate an (anti-aging) repressive chromatin environment.

## 5. Metformin: Restoring the Topography of the Epigenetic Landscape

Aging is reflected by specific DNA methylation changes; almost one third of the CpG (cytosine-guanine dinucleotide) sites show age-associated DNA methylation alterations, of which 60% become hypomethylated and 40% hypermethylated upon aging [165]. The methylation state of select CpG sites can be used to accurately predict the biological age and eventual lifespan, thereby forming the basis of “epigenetic clocks” [166,167,168,169,170,171]. Cell-intrinsic manipulations such as partial reprogramming using short-term cyclic expression of the so-called “Yamanaka factors” (Oct4, Sox2, Klf4, and c-Myc) resets DNA methylation epigenetic clocks and ameliorates cellular and physiological hallmarks of aging while maintaining cell identity [7,8,172]. Metformin might operate as a cell-extrinsic manipulator capable of ameliorating age-associated phenotypes by epigenetic remodeling. Although preliminary, a small clinical trial in which metformin was one of the three drugs administered to nine volunteers for a year (the other two were human growth hormone and dehydroepiandrosterone) found that metformin could partially reverse epigenetic aging (by an average of 2.5 years) while simultaneously regenerating the thymus [173,174]. Metformin appears to promote a regain of responsiveness to prodifferentiation signals [175,176], which might synergistically operate with rejuvenation therapies [15,177]. Accordingly, metformin treatment has been shown to decrease cellular senescence while lowering the abundance of inflammatory cytokines that are hallmarks of the senescence-associated secretory phenotype (SASP) [108,178]. If metformin can target the senescence epigenome including histone modifications, histone variants, DNA methylation, and changes in 3D genome organization, then new studies should explore the benefits that might arise from combining metformin with senescent cell-targeting senolytics and SASP-targeting senomorphics or via the improvement of immune system functions against senescent cells (immunosurveillance) [179,180,181,182,183,184]. Indeed, given that chronic senescence-associated inflammatory signaling locks cells in highly plastic epigenetic states disabled for reparative differentiation, the anti-senescence activity of metformin might operate as a non-cell autonomous mechanism capable of “unlocking” the aberrant epigenetic plasticity of SASP-damaged aging tissues while simultaneously stimulating differentiation of stem cell-like states to successfully achieve tissue repair or rejuvenation [11].

## 6. Metformin: A Guardian of Cell Identity

Metformin, which has been used clinically for over 60 years as a first-line drug for treating type 2 diabetes owing to its effectiveness, safety, and low cost, is now being considered as a promising geroprotector [185,186,187,188]. There is ever-growing evidence that metformin is a central controller of differentiation and cell fates that could directly regulate the biological machinery of human aging via three- and, perhaps, four-dimensional regulatory changes in (epi)genome architecture [189,190,191,192,193] (Figure 3). It might be argued that the proposed ability of metformin to globally preserve the epigenome architecture, thereby ensuring cell fate commitment and phenotypic outcomes despite the challenging effects of aging noise, might preferentially impact the onset of particular subsets of aging-related pathologies such as cancer. Forthcoming studies should carefully evaluate the helpful or detrimental impact of metformin in the dysregulation of the epigenetic landscape not only in neurodegeneration and dementia but also in other aging-related phenotypes including recovery from metabolic disease or tissue damage, functional decline in strength, cognition, immunity, etc. In the meanwhile, it is the time to draw inspiration from such an incredibly simple biguanide to develop a new family of gerotherapeutics that, like parental metformin, would operate in a multi-faceted manner to halt the erosion of the epigenetic landscape, mitigate the loss of cell fate commitment, delay stochastic/environmental DNA methylation drifts, and alleviate cellular senescence.

## Figures and Tables

**Figure 1 biomolecules-10-00780-f001:**
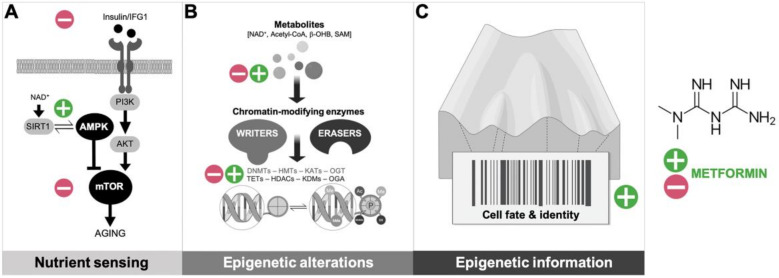
Anti-aging mechanisms of metformin: From metabolic controller of longevity to the sentinel of epigenetic information. (**A**) The insulin/insulin-like growth factor 1 (IGF-1) signaling pathway has prominent aging-modulating effects, which are coupled with the activation status of central nutrient sensors such as sirtuins, AMP-activated protein kinase (AMPK), and mammalian Target Of Rapamycin (mTOR); the ability of metformin to block the insulin/IGF1 signaling while promoting proficient nutrient-sensing pathways (e.g., by activating Sirtuin-1 (SIRT1) and AMPK while suppressing mTOR) can promote organismal metabolic fitness to suppress aging-associated diseases. (**B**) Chromatin-modifying enzymes that catalyze aging-accompanying epigenetic alterations utilizes substrates and cofactors generated by intermediate metabolism; the ability of metformin to regulate the abundance of epimetabolites and/or the activation status of certain epigenetic writers and erasers can directly impact the link between metabolism and the epigenetic control of aging. (**C**) In the context of a reformulated Waddington’s landscape that incorporates the information-theoretic nature of the epigenome [16,17,18,19], the ultimate anti-aging mechanism of metformin might involve the global preservation of the epigenome architecture, thereby ensuring cell fate commitment and phenotypic outcomes despite the challenging effects of aging noise.

**Figure 2 biomolecules-10-00780-f002:**
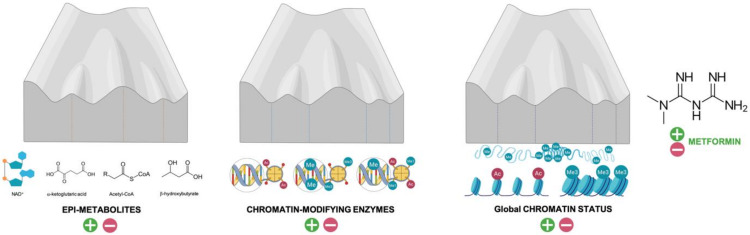
Metformin maintenance of the epigenetic landscape. The height and steepness of hills between two valleys determine, at least in part, the ability of the epigenome to modulate phenotypic plasticity and canalization during differentiation. Metformin might prevent the undesirable interconversion of neighboring cellular epistates occupying adjacent valleys by fine-tuning both the provision of specific epi-metabolites and the activation status of chromatin-modifying enzymes that specify a given cell type, i.e., two key shapers of the Waddington’s epigenomic landscape. As a consequence, metformin might halt the erosion of the epigenetic landscape and delay age-associated epigenetic drift, i.e., the gradual decrease of global DNA methylation leading to the alteration of epigenetic patterns.

**Figure 3 biomolecules-10-00780-f003:**
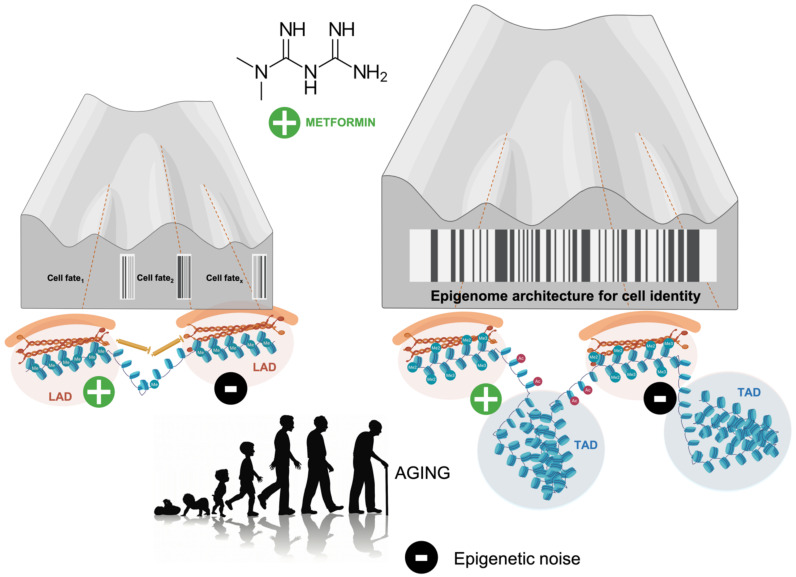
Metformin-regulated noise in the aging epigenetic landscape. Beyond the deterministic model depicted in Figure 2, one should acknowledge that transition states and cell fate decision at branching points of Waddington’s landscape can take place in a discontinuous stochastic manner in which aging noise modulate the probability of transition events. Crucially, the changing deepness and tallness of the valleys and hills can be governed, at least in part, by changes in the three-dimensional (3D) topology of the (epi)genome including lamina-associated domains (LADs) at the nuclear periphery and promoter–enhancer contacts regulating gene expression within topologically-associated domains (TADs) in the nucleus; such structures, which are continually responding to intra- and extracellular cues during development, differentiation, and tissue homeostasis, affects the contour of the epigenetic landscape itself to modulate the effects of noise [16,17,18,19]. Although unproven, the ability of metformin to regulate the 3D shaping of chromatin topology via maintenance of TADs and LADs would favor a physiological/reparative resolution in response to decreased signal-to-noise ratios in cells/tissues undergoing a smoothing of the epigenetic landscape. Perhaps more importantly, the targeting of the cross-talk between (epi)genomic organizers with metformin might provide key mechanistic insights into 4-dimensional (4D) regulatory changes in (epi)genome architecture, where the 4th dimension is aging time.

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
