# Peer review of "Metformin: Sentinel of the Epigenetic Landscapes That Underlie Cell Fate and Identity"

_biomolecules, 2020, doi:10.3390/biom10050780_

Round 1

Reviewer 1 Report

In this paper, the Author reviews the possible effect of metformin as an anti aging compound starting from its well known metabolic effects. The text and conclusions are convincing although largely speculative. The references are complete (sometimes repetitive) but the Author should briefly look at and discuss the data on hexokinase 2 and metformin, which may represent an effective link between metabolic and epigenetic effects of metformin

Patra KC, et al Hexokinase 2 is required for tumor initiation and maintenance and its systemic deletion is therapeutic in mouse models of cancer. Cancer Cell. 2013 Aug 12;24(2):213-228.

Salani B, et al Metformin impairs glucose consumption and survival in Calu-1 cells by direct inhibition of hexokinase-II. Sci Rep. 2013;3:2070. Salani B, et al IGF1 regulates PKM2 function through Akt phosphorylation. Cell Cycle. 2015;14(10):1559-67.

Reviewer 2 Report

Metformin: sentinel of cellular identity

The author has touched a unique but essential concept of how metformin might be able to conserve or repair or manipulate the epigenetic landscape at the genome level in response to assaults caused by various disease conditions. This is a timely and detailed descritption which is necessary to be pusblished. However, I have some minor suggestions,

If the focus of the opinion is the epigenetic landscape in aging, the title needs to be more specific. Something like “ Metformin: Sentinel of cellular identity against aging.”

The author has a unique ability to present thoughts and narrate the importance of the topic in very long, although meaningful sentences. However, if the author seeks the help of English re-writing and breaking down the whole manuscript’s writing in simple and easy to understand-short sentences, the paper will be much more enticing to the readers.

Metformin is a highly versatile molecule, however variable effects of metformin have been observed dependent on multiple factors, such as but not limited to, the target cells, the dose of metformin, duration of metformin treatment, target disease (diabetes, cancer, stroke, dementia), sex, age, etcetera. Hence, it may not be fair to generalize the review of literature only in one direction as the epigenetic landscape protective in nature. If the author has focused opinion mainly on antiaging benefits along with anti-cancer benefits, it might be the case. However, other than these two plausible roles, metformin is helpful or detrimental in other conditions such as brain function, dementia, etc. In those conditions, the expected epigenetic landscape benefit might not be observed similarly. The author needs to address the caveat of his view, adding a section narrating limitations to this approach and how these limitations can be best used to optimize the targets for ultimate benefit.

Abstract: needs complete re-writing to address what is known (premise),  what is presented in the opinion, and conclusion with future direction. The sentences are too long to keep up with the meaning. There are some minor Typographical repetitions “- the ability of ability of”. The addition of a graphical abstract would be very helpful, which summarizes all the figures in the opinion.

Figure one- instead of right/middle/left panel, the author could use A, B, and C labeling for these panels.

Figure 2: What determines the height and steepness hills between valleys in the epigenetic landscape? A slightly easier explanation could make it easy for the readers to understand.
